# Liquid Biopsy in the Management of Breast Cancer Patients: Where Are We Now and Where Are We Going

**DOI:** 10.3390/diagnostics13071241

**Published:** 2023-03-25

**Authors:** Carlotta Mazzitelli, Donatella Santini, Angelo Gianluca Corradini, Claudio Zamagni, Davide Trerè, Lorenzo Montanaro, Mario Taffurelli

**Affiliations:** 1Department of Medical and Surgical Sciences (DIMEC), Alma Mater Studiorum, University of Bologna, 40126 Bologna, Italy; 2Unit of Pathology, IRCCS Azienda Ospedaliero, Universitaria di Bologna, 40138 Bologna, Italy; 3Unit of Oncology, IRCCS Azienda Ospedaliero, Universitaria di Bologna, 40138 Bologna, Italy; 4Departmental Program in Laboratory Medicine, IRCCS Azienda Ospedaliero, Universitaria di Bologna, 40138 Bologna, Italy; 5Unit of Breast Surgery, IRCCS Azienda Ospedaliero, Universitaria di Bologna, 40138 Bologna, Italy

**Keywords:** liquid biopsy, breast cancer, clinical management, circulating tumor DNA, circulating tumor cells

## Abstract

Liquid biopsy (LB) is an emerging diagnostic tool that analyzes biomarkers in the blood (and possibly in other body fluids) to provide information about tumor genetics and response to therapy. This review article provides an overview of LB applications in human cancer with a focus on breast cancer patients. LB methods include circulating tumor cells and cell-free tumor products, such as circulating tumor DNA. LB has shown potential in detecting cancer at an early stage, monitoring tumor progression and recurrence, and predicting patient response to therapy. Several studies have demonstrated its clinical utility in breast cancer patients. However, there are limitations to LB, including the lack of standardized assays and the need for further validation. Future potential applications of LB include identifying the minimal residual disease, early detection of recurrence, and monitoring treatment response in various cancer types. LB represents a promising non-invasive diagnostic tool with potential applications in breast cancer diagnosis, treatment, and management. Further research is necessary to fully understand its clinical utility and overcome its current limitations.

## 1. Introduction

Liquid biopsy (LB) is a minimally invasive approach that has shown, in the last decades, an extremely promising value both at the diagnostic and prognostic level and for monitoring treatment effectiveness across different types of cancer, and analyzing peripheral blood samples or other biological fluids. Unlike conventional tissue biopsy, LB can be performed as frequently as required; it is minimally invasive and has no adverse effects on patients [1].

In comparison with classical tissue biopsy, by means of LB, it is possible to achieve real-time information on tumor status. Despite this, tissue biopsy remains the standard of care for diagnosis due to the reliability of the information provided and the necessity of evaluating numerous biomarkers on tissue sections. However, for advanced neoplastic disorders, LB may play a leading role in situations where it is required to identify potential therapeutic targets [2]. Increasing interest among researchers in LB is motivated by the possibility of obtaining patient- and time-specific information [3]. Moreover, the ease of obtaining biological material for analysis permits to process of serial blood samples in a brief period, that in turn enhances the likelihood of early detection of changes in the patient’s disease status (e.g., disease relapse) [4]. Therefore, in the age of customized medicine, by providing informative and non-invasive diagnostic information, LB is one of the factors facilitating a faster transition toward patient-based targeted treatments. In fact, it can provide useful results for disease screening and for determining the molecular characteristics of the disease [5]. Indeed, unlike tissue biopsy, which is limited in space and time, LB can also be used to mirror the tumoral *heterogeneity* at different times [6]. The concept of tumor heterogeneity delineates the simultaneous existence of cellular subpopulations that are different from each other in both genetic and phenotypic features. This heterogeneity may be present even between the primary tumor and its metastases. This diversity is caused by genetic factors and mechanisms, including clonal selection and adaptive responses [7]. To overcome problems related to the characterization of heterogeneous tumor cellular populations, LB provides the chance to assess the state of cancer at several time points during therapy, with the possibility of driving treatment decisions in real time independently of the site from where the tissue was obtained.

On the practical level, LB can identify in the bloodstream cell-free tumor products (e.g., cell-free DNA, circulating RNA, and circulating tumor DNA–ctDNA) and circulating tumor cells (CTCs) that leaked from the primitive tumor, allowing its molecular characterization, and providing the opportunity for clinicians to adapt personalized treatments to cells resisting emerging therapy [8].

### 1.1. ctDNA

Circulating nucleic acids comprise the fraction of circulating cell-free DNA/RNA originating both from primary and metastatic tumors. This includes short nucleosome-associated fragments (80–200 bp) or longer fragments (>10 kb) encapsulated within extracellular vesicles [9]. ctDNA can be generated by apoptosis, necrosis, or active excretion of tumor cells instead. Apoptosis is the most frequent mechanism involved, and, as a result, released DNA typically measures between 140 and 180 bps in length [10]. The concentration of ctDNA measured in body fluids depends on tumor load and proliferation, and it may reveal alterations and/or aberrations of the genome [11]. ctDNA is rapidly cleared from the blood, and its half-life range is about 2 h, which makes it a dynamic biomarker for monitoring tumor burden [12]. Depending on tumor-specific mutations, the ctDNA may diverge from other non-specific free DNA fragments found in the peripheral bloodstream, which are typically known as free circulating DNA or cell-free DNA (cfDNA), and are derived from non-neoplastic cells. In general, ctDNA is obtained from the blood plasma, and its burden relates to the relative value of ctDNA compared with the total amount of cfDNA that may be poorly represented. The finding that blood-derived cfDNA may represent tumor-specific mutations has supplied a strong basis for the use of ctDNA as a biomarker for clinical application since it can offer information on sub-clonal alterations and clonal heterogeneity in real-time [13]. To achieve this, droplet digital PCR (ddPCR) and next-generation sequencing (NGS) represents very precise and accurate molecular detection methodologies [14]. In particular, NGS provides a comprehensive view of all genetic modification and an overview of all genetic alterations, making it possible to discover unique new specific mutations resulting from the progression of cancer cells under selective pressure induced by therapy. Even if cfDNA can offer worthwhile information for the selection of targeted treatments, these assays do not provide information on protein expression at the same time as the single-cell resolution to characterize genomic heterogeneity, which is indeed possible to obtain from CTCs.

### 1.2. CTCs

Half a century ago, it was demonstrated that CTCs are cancer cells capable of migrating from tumor tissue and merging in the bloodstream, with the possibility of spreading the disease elsewhere [15]. Originally, the initiation of the epithelial-to-mesenchymal-transition (EMT) process led to the exit of cells from the primary tumor and their entrance into the bloodstream [16]. Then, upon entering the blood flow, CTCs are subject to huge stresses and—while most CTCs perish—a subpopulation composed of more aggressive cells manages to survive in circulation. Once in the bloodstream, their half-life is restricted to 2.5 h because of attacks by the immune system. CTCs may then exit the bloodstream arriving at a new place and, after the mesenchymal-to-epithelial-transition (MET), become silents or activate in an extraneous microenvironment to form a metastatic growth. Precisely because CTCs have a critical role in the settlement of metastasis, thought was given to their use in clinical practice. In fact, CTCs can provide a potent tool for monitoring phenotypic changes, tumor progression, and response to therapy. Unfortunately, there are different challenges facing CTCs, including their isolation, detection, and analysis. In regard to CTC isolation, an enrichment method based on physical or biological features is usually used. In detail, the size and density of the cells and the existence of an electrical charge are utilized for physical-based methods. Protein secretion and expression of cell surface antigens are in use for biology instead. To identify and enumerate CTCs, the FDA licensed the CellSearch platform (Menarini Silicon Biosystems) for use clinically in oncologic patients since 2004. Specifically, the test identifies and counts EpCAM-positive, CD45-negative, and cytokeratin 19-positive cells in the bloodstream (cells with an epithelial origin). CTC has been used in clinics for more than 15 years and has been FDA-approved, but despite this, this method of CTC detection raised doubts. Indeed, evidence suggests that a subgroup of CTCs undergoing EMT may evade CTC-specific detection [17]. A cocktail of antibodies consisting of both mesenchymal and epithelial markers can increase the CTC capture effectiveness, but, unfortunately, most detection methods fail to precisely reflect the phenotypical diversity of CTCs during the metastatic waterfall. Another factor that causes false positive results is the contamination of leucocytes, considering that it is very difficult to collect extremely poorly represented CTC from blood cell populations enriched in the white blood cells [18]. In fact, the rareness of CTCs in the bloodstream makes their isolation very difficult: given that leucocytes and CTCs are of comparable size, removal of non-target cells by biochemical features may cause low purity. Since CTC provides a wide variety of analytes and enhanced catch and progress, single-cell analysis methods can also expand our knowledge of CTC, which may help to clarify the metastasis mechanisms. Considering these findings, it could be claimed that the exact detection of all CTCs is vital, and an improved CTC capture platform is required. It is also known that CTCs can flow through the blood independently or in clusters. CTC clusters are composed of primary tumor cells, which are bound by intercellular adherence. Their presence in the blood is very unusual, but they have a greater ability to develop remote metastases compared with individual cells [19]. In fact, cells generate a cluster to protect each other from several stresses that involve immune surveillance and shear force [20]. Usually, considering the total amount of CTCs in the blood, only about 3% circulate as clusters, and in the metastatic patients, in about 50%, a cluster can be detected in the typical 7.5 mL blood samples [21].

Overall, despite the mentioned challenges regarding CTC detection, the prognostic role of CTC has been widely demonstrated by many clinics’ studies. For instance, in a CellSearch analysis, a cut-off of 5 CTCs per 7.5 mL of blood as a measure of poorer outcome is used. The possibility of collecting CTCs in several blood samples over time suggests great potential for CTCs as a dynamic marker for driving real-time drug selection during disease progression. Such cells could raise diagnostic opportunities that rely on the hypothesis that they represent a reflection of metastatic site cell populations that burden tumors [22].

Taken together, ctDNA and CTC analyses are more than just a way to test the disease progression and the effectiveness of treatment and nascent therapy resistance mechanisms, but they also transcend the limits associated with tumor heterogeneity and real-time progression. Therefore, CTCs and ctDNA, used as biomarkers, offer real-time evaluations of tumor trends and constitute a main tool for checking therapy efficiency and choosing the best treatment. Until now, the clinical importance of CTCs and ctDNA biomarkers was mainly evaluated separately, and just a few studies have measured both biomarkers in tandem by using the same blood sample taken from the same patients. Observational clinical studies have already demonstrated that CTCs and ctDNA are appropriate in clinical terms for various types of cancer, but the usefulness of including LB in clinical guidelines in patient care must still be proven [23]. In fact, in many clinical trials, these determinations are only considered in specific tests because what measure LB may eventually replace tissue biopsies needs further clarification. However, ctDNA and CTC could be potentially used as standard biomarkers in the setting of several specific kinds of tumors, among which breast cancer (BC).

Since in this disease, many studies involving the analysis of CTCs or ctDNA have been conducted in the last years, in the next sections, we will focus on the applications of LB in BC patients.

### 1.3. ctDNA and CTC in BC

BC is the most common malignancy in women in the United States, with approximately 87,850 newly diagnosed patients in 2022 [24]. It is predicted that at least 43,250 women are going to die because of this kind of tumor this year, rendering it the second major reason for cancer-related mortality in women. BC is a complicated condition that increases in incidence with age due to accumulating somatic mutations in the breast glands. Breast tissue malignancies are heterogeneous and widely divided into various sub-types depending on molecular aberrations. [25].

In the luminal A subtype, which represents 50–60% of BC patients, the tumors are progesterone receptor (PR) and/or estrogen receptor (ER) positive [26]. Due to a reduced expression of Ki67, a protein correlated to cancer cell proliferation, the luminal A subtype is generally associated with low grades and aggressiveness and a positive prognosis. The luminal B BC subtype is also positive for hormone receptors, with ER and/or PR present but at the elevated expression of Ki67 [27]. For these reasons, the Luminal B patients’ prognosis is marginally worse compared to luminal A patients. Another BC subtype is characterized by the over-expression of human epidermal growth factor receptor 2 (HER2). HER2 is an important molecular marker for BC: about 20% of diagnosed BC patients are HER2-positive, which can be classified as ER^+^ HER2^+^ or ER^-^ HER2^+^ tumors [28]. HER2-overexpressing tumors tend to develop more quickly and may display a worse prognosis. The expressions of ER, PR, HER2, and Ki-67 determine which is the best treatment. In recent years, the exploitation of targeted therapies in BC patients led to an increase in treatment benefits, better prognosis, and higher patient survival for ER^+^ and HER2^+^ tumors. However, the triple-negative breast cancer (TNBC) subtype, described by the no over-expression of HER2, together with the lack of PR and ER expression, is still a clinical challenge due to the absence of peculiar therapeutic targets and the aggressiveness of the disease [29]. Although patients with TNBC get higher rates of complete pathological response with chemotherapy, in comparison with patients with different BC subtypes, their overall survival after chemotherapy is worse than for non-TNBC patients [30]. In case the residual illness remains after the neoadjuvant chemotherapy, TNBC patients are six times more likely to relapse and 12 times more likely to die of metastasis [31].

Analysis of the gene expression pattern of BC datasets discovered TNBC cellular heterogeneity grouped under six molecular subtypes: basal-like (BL1 & BL2), mesenchymal, mesenchymal stem-like, luminal androgen receptor subtype, and immunomodulatory [32]. Chemotherapy remains the standard treatment for patients even if TNBC is divided into six types of tumors. During the chemotherapy, sub-clonal diversity within subtypes contributes to variability in responses and the development of chemoresistance and metastasis [33]. Initially, patients may respond well to chemotherapy. However, under chemotherapy’s selective pressure, a very small number of chemo-resistant cells proliferate and survive after therapy leading to a relapse of the disease [34]. These results indicate the presence of heterogeneity not just spatial (diverse subpopulations in various regions of the tumor) but also temporal (divergences among the primary tumor and its relapse). Thus, it may be considered that the sample procured through a biopsy does not constitute the entirety of the tumor’s composition. The tumor is represented by different tumor cells that diverge in their characteristics and drug sensitivity [35].

Moreover, in BC, an increase in the number of circulating tumor cells (CTCs) and the increment in concentration of circulating tumor DNA (ctDNA) has been brought into correlation with disease development. Therefore, in the BC clinical setting, the analysis of circulating tumor-derived material comes to light as an innovative way of managing patients both advanced and in early BC (EBC). The analysis of circulating tumor-derived analytes (collectively defined as circulome) in BC has mostly concentrated on the qualitative and quantitative characteristics of CTCs and ctDNA [36]. In practice, high concentrations of ctDNA have been associated with a more aggressive and possibly relapsing disease, either in early or advanced settings. Researchers have evaluated ctDNA amounts in patients with EBC to reduce mortality by exploiting an early detection and therapy change. Many studies have evaluated the prognostic value of cfDNA concentrations in BC: it was shown that cfDNA levels rise in patients affected by malignant lesions and are related to tumor size and the clinical stage of lymph node metastasis [37].

As regards CTCs, studies on EBC have shown that these cells can persist after a (neo)adjuvant chemotherapy in many patients. The mechanism that allows these cells to escape cytotoxic treatments is their change in a dormant low-proliferative state for an extended duration, as described before, and could also represent a surrogate marker useful for minimal residual disease (MRD). Moreover, several clinical studies of BC have highlighted the prognostic relevance of CTC’s isolation and characterization, concluding they may be used as follow-up markers and even drive personalized treatment decisions. The idea of substituting tumor tissue biopsies with the aim to achieve therapeutically and diagnostically important information makes CTCs a crucial part of non-invasive “real-time” LB and could be extremely helpful in patients with metastatic breast cancer (MBC) [38].

### 1.4. ctDNA and CTC in MBC

Even though the development of modern medical technology has enhanced the treatment effect of BC, tumor-linked death due to relapse and metastasis continue to be a major clinical problem. To overcome this, CTCs have been proposed as a significant predictor of MBC, and early studies demonstrated their potential to drive therapy decisions [39]. Modifications in the metastases’ biomarker status in comparison with the primary tumor are frequent in MBC. Thus, the molecular characteristics of metastatic lesions must be identified before starting targeted therapy [40]. In fact, it was shown that the number of detectable CTCs in the blood of EBC and MBC patients is negatively associated with progression-free survival. Elevated CTC number correlate with shortened progression-free survival (PFS) and overall survival (OS) in MBC, even if CTC enumeration is also a prognostic factor in nonmetastatic BC [41]. The next step in clinical implementation involves demonstrating the utility of CTC as an LB biomarker.

A great deal of research has been done on the utility of CTCs and ctDNA in BC for driving treatment decisions, resulting in the first liquid biopsy-based FDA approval for MBC in 2019. 

Overall, the usefulness in clinical practice was proved for three cfDNA-based tests already approved by the U.S. Food and Drug Administration (FDA): the therascreen PIK3CA RGQ PCR kit, the Epi proColon test and the cobas EGFR Mutation Test v2 [42]. Moreover, as already mentioned, in the CellSearch system, the FDA approved the CellSearch analysis for CTCs detection in patients affected by MBC.

## 2. Where Are We Now

To provide the reader with actual and timely examples of applications of LB in breast cancer, here we report a list of the major clinical trials using LB (CTC and ctDNA) as a biomarker in this disorder which are ongoing or concluded in the last 5 years (see Table 1 for a summary).

### 2.1. EBC

#### 2.1.1. ctDNA

A considerable number of methods have been utilized in recent years to identify and measure ctDNA in EBC patients. For example, the plasma of these patients on matched pre- and post-surgery samples was performed using highly sensitive digital droplet PCR for the detection of PIK3CA mutations in ctDNA. These examinations clarified the relevance of ctDNA in checking tumor mass and monitoring the development of resistant clones to allow adequate treatment.

Moreover, the *NeoALTTO* trial exanimated at the correlation between ctDNA and reaction to anti-HER2 treatment and found that HER-2-positive tumors without ctDNA at baseline showed better rates of pathologic complete response (pCR), suggesting that ctDNA may be exploited as a new biomarker for neo-adjuvant chemotherapy (NACT) reaction in HER-2-positive BC. In this context, the genomic variations of ctDNA for their potential utility as biomarkers for EBC treatment have been investigated [43]. The sequencing of ctDNA and of corresponding samples of tissue displayed that intra-tumoral heterogeneity discovered in tissues of the tumor was mirrored in ctDNA data. Additionally, high ctDNA concentrations after surgery have been associated with a greater percentage of lymph node metastasized, suggesting the possibility of relapse and remote metastasis.

Another study verified the effectiveness of ctDNA in predicting recurrence in triple-negative breast cancer (TNBC) patients still with the disease after NACT [44]. The ctDNA analysis showed that the relapse in these patients could be predicted with elevated specificity but minor sensitivity. Moreover, relapse was rapid during the detection of ctDNA. Diagnostic assays used for an accurate prediction of residual disease after NACT are required in BC. Response to NACT can drive therapeutic choices, including surgical removal and radiation therapy. Due to the low sensitivity of traditional diagnostic techniques, a biomarker of therapeutic response able to specifically distinguish the remaining disease from the removed disease would enable patients to achieve the more appropriate therapy [45]. 

Finally, a patient-specific multiplexed cancer mutation analysis method for targeted digital sequencing. (TARDIS) was programmed [46]. This application made it possible to detect tiny quantities of residual DNA in patients’ plasma. Patients displaying a pCR had reduced ctDNA concentrations compared to those with the remaining disease. 

Overall, these results showed that serial ctDNA measurements could be analyzed for the characterization of patients at risk of recurrence.

#### 2.1.2. CTCs

For HER2-positive patients with primary BC, NACT that included trastuzumab was evaluated in *GeparQuattro* clinical study [47] utilizing the CellSearch assay for CTC isolation was FDA-approved. CTC numbers were low in early-stage disease, and, in spite of a reduction in CTC numbers after NACT, a correlation between persistent CTC numbers and treatment response was not verified. HER2 immunoscoring of CTCs was designed for patients having CTCs overexpressing HER2, with the aim of appropriately delivering HER2-targeted treatments.

A further study examined the prognostic relevance of CTC positive to cytokeratin (CK-19)- in EBC after NACT and identified this as an autonomous risk factor [48].

Moreover, the possible role of CTCs in operable BC was studied, showing that CTCs were present either before or after the surgery in 30% of patients. It was discovered that this persistence of CTCs in NACT, noting a patient sub-population, correlated with an increased chance of relapse. It was determined that CTCs could prognosticate early metastatic recurrence after NACT in surgically resectable and BC in advanced status. In fact, in stage I-III TNBC, the detection of CTCs after NACT displays a number ≥1 CTCs as predictive of relapse [49].

Moreover, the *BEVERLY-2* trial on primary BC patients demonstrated the independent prognostic potential of CTC number before and after NACT. This prospective study examined the safety and effectiveness of NACT after a treatment using bevacizumab and trastuzumab in patients with HER2-positive BC. Following the *BEVERLY* trial, the implication of CTCs in cancer dissemination and their possible use for BC patients’ subpopulations was clearly constituted [50].

Several additional studies focused attention on determining the potential role of CTC as a biomarker in BC. In one of these studies, it was shown that the expression of PR, ER, and Ki-67 in BC may be properly identified on CTCs in real-time, giving guidance for cancer monitoring and therapy. In all, 198 HER2-positive BC patients have been enrolled, and the CTC detection rate was 79.8% (158/198): >30% of CTCs were strongly positive for HER2, however just 41.1% (65/158) patients had consistent histology and CTC HER2 status, along with other 58.9% (93/158) patients displaying HER2 positive histology, and CTC HER2 status negative; 98 (62.0%) patients were positive for estrogen receptor (ER) while 75 (47.5%) patients were positive for progesterone receptor (PR); the Ki-67 positive level was higher than 50% in 88 (55.7%) patients. In summary, this study proved the importance of CTC HER2 real-time screening in HER2-positive BC patients [51].

Moreover, the *IMENEO* study, currently performed on more than 2000 BC patients treated with NACT, aims to evaluate the clinical utility of CTC as a prognostic marker. It was verified that the statistical relevance of CTC enumeration rose with the number of CTCs, from 0 to 1 cell and an HR of 6.25 (95%) in five or even more cells, strengthening the idea that CTC enumeration may be a marker with quantitative quality.

Consistently, in the *E5103* trial, the role of CTC enumeration to predict the high-risk hormone receptor-positive HER2-negative BC tardive relapse 5 years after diagnosis is currently under examination. In this study, the recurrence rates in patients positive for CTCs were 21.4% per person per year after a follow-up vs. 2.0% in patients negative for CTCs. In this study, the association of CTC presence confirms its predictive role on disease recurrence.

In addition, in the *SUCCESS-A* trial, the identification of CTCs is confirmed as a risk factor for an increased risk of death and relapse after two years of follow-up. Considering a follow-up of 5 years in patients with a well-known CTC number, the trial demonstrated that positive CTCs were discovered in 7.8% of patients and were linked to an increase in relapse. It has also been indicated that patients that were under the cut-off of CTC (≤1) had a much longer life expectancy. Therefore, CTC count can represent a useful tool for advanced disease staging and stratification [52].

A different approach relied on the combined PGK1/G6PD (GM) markers expression and classified CTCs in different metabolic subtypes. Cells were marked by DAPI^+^CD45^−^GM^+^ and DAPI^+^CD45^−^GM^−^ and respectively distinguished as GM^+^CTCs and GM−CTCs. GM^+^CTCs concentrations clearly rose in the group of metastatic in comparison with the nonmetastatic one (*p* < 0.001). Utilizing a limit of CTCs ≥ 3/5 mL and GM^+^CTCs ≥ 2/5 mL, the beneficial performance of GM^+^CTCs and CTCs in BC metastasis diagnosis was tested. The data confirmed the hypothesis that metabolically active CTCs might constitute an aggressive subpopulation with a role in the development of metastasis. Therefore GM^+^CTCs were discovered as a more specific marker than simple CTCs detection, and CTCs marked by PGK/G6PD^+^ could represent another important biomarker for prognosis prediction and metastasis diagnosis in BC patients [53].

To test a new therapeutic strategy for TNBC patients’ management, 32 patients in all with TNBC were enrolled for a prospective study. The CTCs enumeration was counted in 7.5 mL of whole blood by using CellSearch. EpCAM and CKs positive CTCs were identified in 14 patients, and, significantly, at the time of sample collection, all the patients with positive CTC enumeration were metastatic, with eight of these that showed ≥5 CTCs. In addition, patients with CTCs ≥5 displayed a considerably shorter PFS and OS compared with the ones with <5 CTCs, highlighting the prognostic potential of the EpCAM^+^ CTC population in patients with TNBC [54].

Overall, these studies demonstrated that CTC enumeration and, even more notably, their characterization represents an optimal option to achieve important information to establish novel diagnostic, prognostic, and monitoring alternative for patients affected by EBC.

### 2.2. MBC

#### 2.2.1. ctDNA

In MBC, LB can provide evidence on the genomic situation of the tumor subpopulations and even on the burden of the tumor.

Different studies investigated the prognostic relevance of ctDNA in MBC. For example, the identification of ctDNA on 640 patients affected by different tumor types using the ddPCR method discovered that ctDNA numbers were >75% higher in MBC patients and 50% higher in localized breast adenocarcinoma patients [55]. Moreover, a meta-analysis of ten available studies was conducted to determine the correlation between cfDNA and OS. This analysis established a solid link between cfDNA and OS and disease (and recurrence) free survival.

On the same topic, in the BEECH trial, an early ctDNA trend was established to predict the PFS in MBC. This trial tested the effectiveness of capivasertib, an AKT inhibitor, in combination with paclitaxel, that is already in use as the first-line chemotherapy in MBC that was HER2^+^ or HER2^−^ and harbored PIK3CA mutations. The results highlighted that ctDNA dynamics at the start of processing might be used as a marker for PFS and an early predictor of treatment efficacy. These results show that the development of therapies acting on cancer cell adaptability could play a role in blocking their intrinsic resistance, improving the survival of patients with late-stage disease [56].

Overall, these findings showed the prognostic and predictive potential of ctDNA in MBC.

#### 2.2.2. CTCs

##### Prognostic Relevance of CTCs

In EBC, CTCs may be discovered in a large number of patients and are considered to represent a marker for potential precursors and residual disease of consequent metastatic illness.

In MBC, the detection of CTCs has proven to be a reliable, independent prognostic factor. In the early setting of disease, the cut-off is set up on ≤1 CTC in order to discriminate among patients with good and bad prognoses, while ≤5 CTCs are set up in advanced disease per 7.5 mL of blood. In 2020, the DETECT trials, presented at the San Antonio Breast Cancer Symposium, described that blood samples from MBC patients for participation in *DETECT III* and *IV* trials were examined. In particular, *DETECT* studies represented the largest study program on CTC-based therapy from around the world and started to clarify the meaning of characterizing the CTC and its exploitation in targeted therapy. The trials’ goal was to evaluate the safety of treatment with dual HER2-targeted therapy (trastuzumab and pertuzumab) together with chemotherapy or endocrine therapy. This trial was the first that enrolled patients relying on their CTCs HER2-phenotype. In fact, the patients’ selection was based on the HER2 status of the primary tumor. Consequently, they were sorted into divergent trials of DETECT, and the CTCs and PFS clearance could predict clinical effectiveness. Examination of patients selected for these studies proved the strong prognostic value of CTC enumeration in MBC [57]. Intriguingly, either the 1 CTC (BC) or the 5 CTCs (MBC) cut-offs were remarkably correlated to OS.

In a recent study, the expression of ER in CTCs was evaluated in sixty patients. CTCs were identified in 50% of ER^+^ MBC patients: many of those patients had ER^−^ CTCs, while a third expressed a mix of ER^+/−^ CTCs. In detail, fourteen patients underwent progression, while ten patients stayed stable or even had disease regression. Interestingly, patients under progression exhibit a transition from CTC negative to CTC positive or displayed an increase in the number of CTCs. After replacing endocrine therapy with chemotherapy, a reduction in CTC enumeration was obtained. The reduction of CTC number from the baseline until the end of therapy was substantial in most samples as well as the conversion in CTC-negative status. Since these results indicate that a reduction or a rise of CTCs number is related to stable disease or progression, respectively, the distinctions between the two blood samples of the two clinical groups of response were confronted. It was found that an increase in the CTC number was much greater in blood samples from patients with a progression of disease than those from patients with stable disease. These data indicated that the rise in the CTC numbers is strongly associated with worse overall survival, also highlighting a significant correlation between the increase of CTCs and progression-free survival. Therefore, monitoring ER-CTC status could be added to CTCs enumeration as prognostic value and may be useful in therapy resistance prediction.

##### CTCs for Therapy Monitoring

CTCs enumeration was the first clinically dedicated liquid biopsy biomarker, and although the prognostic implications were consistently proven, its role in the clinical decision-making process has been debated. In particular, the *SWOG 0500* phase III trial was the first to aim at using CTCs as a clinical decision tool. In this study, CTCs were measured in 595 MBC patients who had not yet begun the first-line therapy. Importantly, patients with limited CTC numbers at baseline showed the most favorable clinical results.

*CirCe01* trial (NCT01349842) was another study investigating CTC-based therapy monitoring. Here the suggested early chemotherapy changes in patients without CTC response culminated in a longer median OS and PFS [58].

##### Selection of Therapy CTC-Driven

French *STIC* CTC represents the first trial based on liquid biopsy finding to prove that CTCs’ numbers could drive treatment decisions in MBC.

In detail, STIC examined the ideal first-line therapy for HER2^−^ MBC patients. Following randomization, patients who were in the control arm received a therapy based on a medic’s decision: endocrine therapy if the tumor was categorized as clinically low risk or chemotherapy in the case of a high-risk of disease. In regards to the CTC arm, the therapy was selected in line with the results obtained through the CellSearch analysis: patients with <5 CTCs per 7.5 mL blood were treated with endocrine therapy, while those with ≥5 CTCs with chemotherapy. After several months of follow-up, the OS and PFS in either group were equal. Intriguingly, patients in discordant status (e.g., clinically low-risk and ≥5 CTCs or clinically high-risk but with <5 CTCs) received a benefit from chemotherapy; namely, both the OS and PSF were consistently prolonged in patients under chemotherapy. Therefore, it may be highlighted that the CTC-based therapy decision brought greater clinical results for the first time [59].

##### CTCs and ctDNA

In MBC, studies have been performed evaluating the utility of CTCs and ctDNA in the same patient’s sample for driving therapy choices.

For example, a study proved that the tissue features were reflected by the mutational status of ctDNA, indicating that biomarkers in circulation could uncover supplementary information. In fact, it is possible to achieve information on mutational profiles from ctDNA and CTCs at the same time, but while ctDNA represents a more frequent event and it is easier to analyze, CTCs could represent a more specific auxiliary biological resource useful for tumor heterogeneity characterization. The tumor genesis of diverse CTC sub-populations through mutational assessment has been proven. Therefore, if verified in a wider population, the analysis of sub-types of CTCs may find application in MBC to design fuller disease imagery.

Moreover, in *ELIMA*’s study, a multi-parametric approach was chosen for the analysis of CTC gDNA, mRNA, CTC mRNA, and cfDNA from blood-depleted CTCs. A reduction in the ctDNA amount and CTC count from baseline through the second cycle of paclitaxel and bevacizumab in HER2−MBC patients was evaluated and resulted in an independent prognostic value. Importantly, alternative analysis in CTCs was proven to be useful in identifying new resistance mutations in comparison to cfDNA, highlighting the potential advantage of DNA analysis in CTCs compared to cfDNA. Therefore, CTC gDNA, CTC mRNA, EV mRNA, and cfDNA could offer complementary information, and a multi-parametric liquid biopsy method may represent a good strategy for the clinical practice because of its ability to provide information on the transcriptomic and disease complicatedness [60].

Furthermore, in the *COMET* trial, the analyses were conducted on 198 patients in which ctDNA concentration and CTC enumeration were attainable at baseline. It has been shown that CTC numbers were superior in patients with metastases. In a preliminary evaluation, if exanimated separately, CTCs and ctDNA levels after four weeks had a significant prognostic value on either OS or PFS. They also resulted in independent factors related to low PFS and OS in multivariable analysis. This suggests that the combination of ctDNA and CTCs may raise the range of patients evaluable for marker monitoring and detection [61].

Finally, a study involving 98 patients with advanced or MBC demonstrated that those with at least one CTC had poorer survival compared with those with no CTCs, providing useful prognostic information [62].

These results may support the use of CTC enumeration as a stratification tool for patients in need of specific therapy. Moreover, CTC counts at the time of progression may give important prognostic information for the post-therapy outcome, sustaining other studies on the role of CTCs in managing treatment decisions in this setting.

## 3. Where Are We Going

In this review, open questions involving the applications of LB in the management of BC are discussed. In summary, LB turned out to be useful at different disease stages and for different purposes, including prognostic assessment, evaluation of the response to treatment, and choice of treatment. One important limitation in the assessment of the role of LB in EBC is represented by the long follow-up time required to draw clinically relevant conclusions. In this sense, the evaluation of surrogate early response markers can be considered an important tool to shorten the observation time required. For instance, in neo-adjuvant chemotherapy routinely performed in HER2^+^ and TNBC, the evaluation of pathological response after surgery can provide a very relevant characterization of the response to treatment related to patient outcome. The combination of early indicators of response can significantly shorten the time needed to evaluate CTC and or ctDNA significance in EBC. This is the case of an ongoing study performed at our institution in which CTCs are evaluated before and after neo-adjuvant treatment in HER2^+^ or TNBC, and the results are related to pathological response after surgery. The results of studies employing early surrogate response markers can shorten the time for the evaluation of the usefulness of LB in EBC, possibly leading to its application into clinical practice (e.g., in the definition of more or less aggressive therapeutic protocols).

Concerning MBC, considering the higher number of CTCs that are commonly detected, the standardization of the method still represents an issue. In fact, when dealing with an advanced tumor, a considerable number of cells can fall outside the current “standard” criteria used to define CTCs (i.e., being ep-CAM/CK positive and CD45 negative). In this sense, the definition of additional recognized markers of CTCs could help the recognition of the method at the clinical level. Figure 1 summarizes the different potential applications of LB in BC and the related issues that are still to be solved.

## Figures and Tables

**Figure 1 diagnostics-13-01241-f001:**
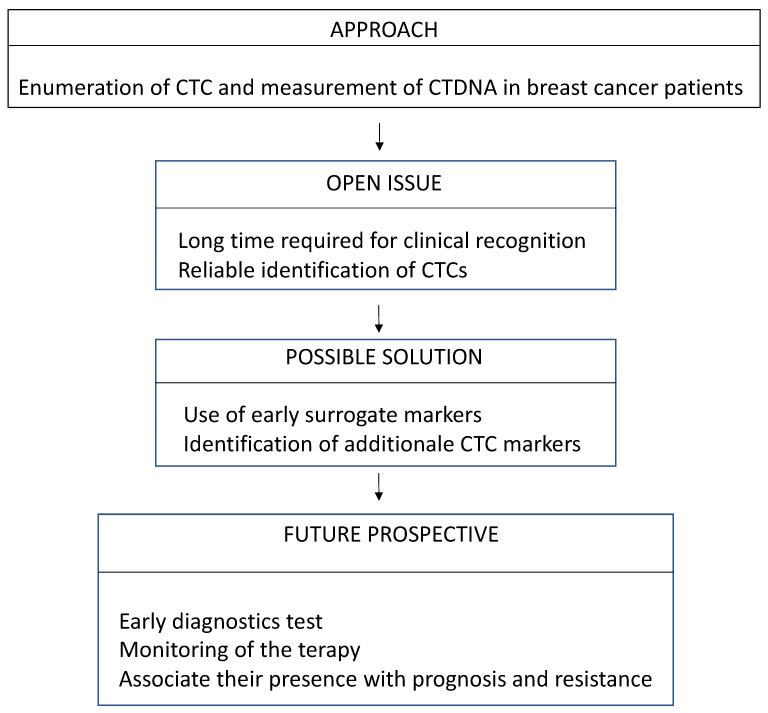
Pitfalls and solutions: The limitations of liquid biopsy in breast cancer detection and monitoring.

**Table 1 diagnostics-13-01241-t001:** List of the mentioned clinical trials. A search for the name of the trial provides additional information at www.clinicaltrials.gov (accessed on 31 January 2023).

**EBC**		
**LB Marker**	**Trial**	**Study Design**
ctDNA	*NeoALTTO*	Biomarker for NACT
CTC	*GeparQuattro* *IMENEO*	Count before and after NACT
	*BEVERLY-2* *E5103*	Enumeration as a high-risk predictor of late recurrences
	*SUCCESS-A*	Detection as a risk factor of death after the follow-up
**MBC**		
**LB Marker**	**Trial**	**Study Design**
ctDNA	*BEECH*	Amount as a predictor for PFS
CTC	*DETECT*	Exploitation in targeted therapy
	*SWOG 0500* *CirCe01*	Clinical decision tool for therapy monitoring
CTC and ctDNA	*ELIMA*	Complementary information on the genomic and transcriptomic disease complexity
	*COMET*	Significant prognostic value on both PFS and OS

## Data Availability

Not applicable.

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
