# Peer review of "Liquid Biopsy in the Management of Breast Cancer Patients: Where Are We Now and Where Are We Going"

_diagnostics, 2023, doi:10.3390/diagnostics13071241_

Round 1

Reviewer 1 Report

Comments and Suggestions for Authors

1.     Method: is this a proper review? selective review? Please provide a methods section.

2.     I recommend correcting (rewriting) the summary of the article which will reflect the main idea of the review and the authors' own thoughts.

3.     Concerning the citations/references: in some parts of the text the provided content needs more references, e.g. lines 51-59.

4.     In my opinion the topic is interesting, however, it is described too general. The details are required to be presented, because the topic of liquid biopsy is well known so far. Basing on experience gained on the other cancer analysis, the additional conclusions could be formed. The authors do not indicate what problems exist for the widespread use of liquid biopsy in a clinical practice. Moreover, the additional tables and figures can increase paper quality. 

Author Response

1.     Method: is this a proper review? selective review? Please provide a methods section. 

A: This is a traditional review manuscript, in general the mentioned references and works have been selected after an informed review of the literature by the authors and do not derive from systematic search of the literature. However, the last section derives from a search focused on studies identified by the following keywords: Liquid biopsy and breast cancer and (circulating tumor cells or circulating tumor DNA), the search was limited to the last 5 years (2017-2023). This information was added before the second section of the review (Where are we now?) 

2.     I recommend correcting (rewriting) the summary of the article which will reflect the main idea of the review and the authors' own thoughts. 

 A: the article abstract has been modified according to the reviewer’s suggestion 

3.     Concerning the citations/references: in some parts of the text the provided content needs more references, e.g. lines 51-59.

A: We have included the required references

4.     In my opinion the topic is interesting, however, it is described too general. The details are required to be presented, because the topic of liquid biopsy is well known so far. Basing on experience gained on the other cancer analysis, the additional conclusions could be formed. The authors do not indicate what problems exist for the widespread use of liquid biopsy in a clinical practice. Moreover, the additional tables and figures can increase paper quality.  

A:  line with this Reviewer’s comment we expanded the conclusion section dealing with existing problems related to liquid biopsy application, also adding a summarizing scheme.   

Reviewer 2 Report

The review is comprehensive and well-written, but minor corrections are required

L174-175

the luminal  A subtype is generally associated with lower grades, aggressiveness, and positive prog nosis.

Apparently, should be – “low grades and aggressiveness, and positive prognosis”

L177 - elevated concentration of Ki67 –

Better “expression of Ki67”

L197-198

Chemotherapy remains the standard of treatment for patients even if TNBC is collected into six different tumors.

Should be “divided into six types of tumors”

L198-199

During the chemotherapy sub-clonal variety among subtypes help with the variability in development and response to chemoresistance and metastasis .

Unclear

L238-242

In fact, it has been proven that the  number of detected CTCs in the bloodstream of EBC and MBC patients is badly correlated  with progression-free survival. High number of CTC correspond to a shorter progression free survival (PFS) and overall survival (OS) in MBC, even though CTC count is a prognostically important factor even in non-metastatic BC.

Unclear

L309

examined the security and effectiveness of NACT

Apparently, should be “safety”

L438

In detail, STIC examined the ideal first-line therapy for HER+ or HER2- MBC patients

According to ref 57, only Her2- patients were considered

Author Response

1. L174-175 

the luminal  A subtype is generally associated with lower grades, aggressiveness, and positive prog nosis. 

Apparently, should be – “low grades and aggressiveness, and positive prognosis” 

A: We corrected the text as suggested 

2. L177 - elevated concentration of Ki67 – 

Better “expression of Ki67” 

A: Modified as requested 

3: L197-198 

Chemotherapy remains the standard of treatment for patients even if TNBC is collected into six different tumors. 

Should be “divided into six types of tumors” 

  A: Modifications done according to the request 

4. L198-199 

During the chemotherapy sub-clonal variety among subtypes help with the variability in development and response to chemoresistance and metastasis . 

Unclear 

A: We modified this sentence to make it more clear 

5. L238-242 

In fact, it has been proven that the  number of detected CTCs in the bloodstream of EBC and MBC patients is badly correlated  with progression-free survival. High number of CTC correspond to a shorter progression free survival (PFS) and overall survival (OS) in MBC, even though CTC count is a prognostically important factor even in non-metastatic BC. 

Unclear 

A: We completely rewrote the sentence to render it clearer

6. L309 

examined the security and effectiveness of NACT 

Apparently, should be “safety” 

A: Modified as indicated 

7. L438 

In detail, STIC examined the ideal first-line therapy for HER+ or HER2- MBC patients 

According to ref 57, only Her2- patients were considered 

  A: Corrected. We apologize for the mistake  

Round 2

Reviewer 1 Report

1.  This is a traditional review manuscript, in general the mentioned references and works have been selected after an informed review of the literature by the authors and do not derive from systematic search of the literature. However, the last section derives from a search focused on studies identified by the following keywords: Liquid biopsy and breast cancer and (circulating tumor cells or circulating tumor DNA), the search was limited to the last 5 years (2017-2023). This information was added before the second section of the review (Where are we now?) 

Аnswer accepted

2.   Аnswer accepted

3. Аnswer accepted

4. Аnswer accepted